# Purity, Danger, and Patriotism: The Struggle for a Veteran Home during the COVID-19 Pandemic

**DOI:** 10.3390/pathogens12030482

**Published:** 2023-03-18

**Authors:** Ippolytos Kalofonos, Matthew McCoy

**Affiliations:** 1HSR&D Center for the Study of Healthcare Innovation, Implementation & Policy (CSHIIP), VA Greater Los Angeles Healthcare System, Los Angeles, CA 90073, USA; 2UCLA/VA Center for Excellence for Veteran Resilience and Recovery in Homelessness and Behavioral Health, Los Angeles, CA 90073, USA; 3Center for Social Medicine and Humanities, Jane and Terry Semel Institute for Neuroscience and Human Behavior, David Geffen School of Medicine, UCLA, Los Angeles, CA 90095, USA; 4UCLA International Institute, 11248 Bunche Hall, Los Angeles, CA 90095, USA

**Keywords:** COVID-19, homelessness, Veteran, encampment, syndemic

## Abstract

The coronavirus disease 2019 (COVID-19) pandemic rendered congregate shelter settings high risk, creating vulnerability for people experiencing homelessness (PEH). This study employed participant observation and interviews over 16 months in two Veteran encampments, one located on the grounds of the West Los Angeles Veteran Affairs Medical Center (WLAVA) serving as an emergency COVID-19 mitigation measure, and the other outside the WLAVA gates protesting the lack of onsite VA housing. Study participants included Veterans and VA personnel. Data were analyzed using grounded theory, accompanied by social theories of syndemics, purity, danger, and home. The study reveals that Veterans conceptualized home not merely as physical shelter but as encompassing a sense of inclusion and belonging. They sought a Veteran-run collective with a harm reduction approach to substance use, onsite healthcare, and inclusive terms (e.g., no sobriety requirements, curfews, mandatory treatment, or limited lengths of stay). The twin encampments created distinct forms of community and care that protected Veterans from COVID-19 infection and bolstered collective survival. The study concludes that PEH constitute and belong to communities that provide substantial benefits even while amplifying certain harms. Housing interventions must consider how unhoused individuals become, or fail to become, integrate into various communities, and foster therapeutic community connections.

## 1. Introduction

The coronavirus disease 2019 (COVID-19) pandemic rendered congregate shelter settings high risk, creating vulnerability for people experiencing homelessness [1,2,3]. Residing outdoors as opposed to indoors was protective with respect to COVID-19 transmission [4]. The COVID-19 pandemic arrived as Los Angeles County (LAC) faced a housing crisis with 64,000 people, including 4000 Veterans, experiencing homelessness [5]. Given the risks of COVID-19 transmission in indoor settings and fears that homeless individuals might contract and spread the virus, LAC instituted temporary emergency mitigation measures, including offering hotel and motel vouchers and pausing forced removals of homeless encampments known as “sweeps” [6]. The West Los Angeles VA (WLAVA) took the unprecedented step of operating an outdoor tent encampment for unhoused Veterans on the health center campus [7]. Concurrently, the moratorium on encampment sweeps enabled a group of Veterans sleeping just outside the WLAVA gates to develop a more permanent settlement. COVID-19 facilitated a natural experiment in responding to the existing crisis of Veteran homelessness: two encampments beside one another, each providing unhoused Veterans shelter and care, one operated by the VA and the other run by the Veterans themselves. 

### 1.1. Background

This article addresses interacting biological and social pathogens. While the SARS-CoV-2 virus is the biological pathogen causing the COVID-19 pandemic, homelessness can be conceptualized as a social pathogen. This notion of “social pathogen” is influenced by anthropologist Mary Douglas’ classic work on pollution and taboo entitled Purity and Danger, where she defines “dirt” as “matter out of place” [8]. Dirt, and related ideas of pollution and defilement, is defined as something that disrupts or violates the established social order. Due to widespread acceptance of the role of microorganisms in the transmission of disease, contemporary ideas of dirt are dominated by ideas of pathogenicity and hygiene. People experiencing homelessness (PEH) disrupt the accepted order by residing in spaces not intended for human habitation. They are seen to embody “dirt”—disorder and pathogenicity—in their soiled clothes and battered bodies. They are feared as [social] pathogens and as sources of disease, crime, and vice. As a result, PEH are profoundly marginalized and victimized by neglect and violent crime at staggering rates [9,10].

Due to their multiple and intersecting socioenvironmental vulnerabilities, PEH have higher rates of mortality, physical illness, mental illness, and substance use disorder (SUD) and are more likely to utilize emergency departments, be admitted to the hospital at a younger age, and have longer lengths of stay at a greater cost as compared to the housed population [11,12]. PEH have increased rates of infectious diseases such as HIV, tuberculosis, and community-acquired pneumonia; chronic medical conditions such as chronic obstructive lung disease, diabetes, and cardiovascular disease; mental illnesses including psychotic disorders and substance abuse [11]. Many of these illnesses substantially increase susceptibility to COVID-19, and after contracting COVID-19, PEH are more likely to require ICU stays and have higher mortality rates [13].

Scholars have proposed the term “syndemic” to describe the intersecting risk factors for COVID-19 and homelessness, rejecting as reductive the biomedical-risk-factor and lifestyle models routinely applied to the analysis of chronic illness and infectious disease [14]. A syndemic is “the adverse synergistic interaction of two or more diseases or other health conditions promoted or facilitated by social and environmental conditions” [14] (p. 7). While homelessness is not itself a disease, it is a condition that creates susceptibility to multiple diseases that interact with COVID-19 to increase morbidity and mortality. The chronic inflammation linked to many of these disorders is augmented by the acute inflammation of SARS-CoV-2, leading to an intense immunological reaction that can result in multiple organ failure [15]. The syndemic nature of the COVID-19 pandemic in the United States highlights the ways infectious disease risk is mediated and magnified by socioenvironmental conditions in highly unequal ways [16,17,18,19,20].

This article examines twin encampments. The first, an emergency COVID-19 mitigation intervention for unhoused Veterans called the “Care, Treatment, and Rehabilitation Service” (CTRS), was a “sanctioned encampment” on the grounds of the WLAVA. Beginning as 25 tents in a VA parking lot in April 2020, CTRS infrastructure continuously improved, increasing capacity to 100 tents by September 2020 (further described below). Though CTRS staffing varied, full-time staff included two social workers and two Veteran peer specialists. Supplemental/occasional staff included additional social workers, a preventative medicine physician, an occupational therapist, and primary care clinicians providing bimonthly onsite urgent care. Veterans had access to regular healthcare via virtual or in-person appointments within the campus medical center (½ mile walk). 

The second encampment consisted of as many as 50 Veterans camping in a row on the sidewalk abutting the WLAVA fence. They leveraged the social anxiety around the pandemic to render visible the plight of the homeless Veterans. Simultaneously displaying their “patriotism,” “ingenuity,” and “vulnerability” through their encampment, these Veterans argued they deserved the home on the VA campus that had been promised in the original land grant of 1888. This “unsanctioned” encampment became known as “Veterans Row.” Veterans on Veterans Row instigated a debate within the VA and in the media about what constituted a home for Veterans amid a global pandemic and in a local context of stark inequities in Brentwood, one of the most affluent neighborhoods in the country. Through this public debate, Veterans challenged the VA to collaborate with them in providing better housing options for Veterans on the VA campus in response to the crises of COVID-19 and Veteran homelessness.

Veterans argued that what they needed to mitigate their vulnerability to COVID-19 infection was neither solely social distancing, masks, and vaccines nor a motel or apartment voucher, but rather, a home for Veterans on the VA campus. This ethnographic research reveals that Veterans conceptualized “home” not merely as physical shelter but as constituted by “feelings, practices, and relationships [with others in] familiar spaces” that produce “a powerful sense of belonging” [21]. Understood in this way, home is also “contestable and fragile” and “a domain not only of belonging but also of potential alienation when attempts to make home fail or are subverted” [21]. Unhoused Veterans lacked a home not simply because they lacked shelter: they lacked a sense of inclusion and belonging and felt abandoned by their country and fellow citizens. In the context of protracted displacement like this, home comes to be understood as a set of homemaking practices, strategies, and routines, even amid temporary dwellings [22,23]. While Veterans residing in Veterans Row and CTRS did not “theorize” home explicitly, their demand for a Veteran-run collective where they could live together, with a harm reduction approach to substance use, compassionate onsite healthcare, and inclusive terms (e.g., no sobriety requirements, curfews, mandatory treatment, or limited lengths of stay), provided an implicit model of a viable home for them during the syndemic. 

While “home” can have a material and physical designation, it can also be understood as an ongoing process of negotiation and as a set of practices that draw from experience and work towards an ideal future [21] (p. 9) [24]. Home can be a space of belonging, yet it can also be a site of alienation, exclusion, instability, or a desire for something else [21] (p. 12). Home evokes nostalgia, yet “even its most altruistic and successful version exerts a tyrannous control over mind and body” [25] (p. 303). Contexts such as encampments also have their own forms of “homeless heritage” that accrue meaning through repeated interactions with the material and social aspects of the place over time [26]. In the natural experiment described here, each encampment featured distinct benefits and costs and served different purposes for different actors at different times. What it meant to experience safety, community, and solidarity in a pandemic was debated and experimented with within each encampment and between the two, with neither approach always serving everyone. 

### 1.2. WLAVA: Soldier’s Home or Healthcare System?

The question of home, and not simply shelter, has been at the forefront of Veteran activism at WLAVA. The WLAVA stands on 388 acres, surrounded by West LA, Westwood, and Brentwood, neighborhoods with median home values of $1.5 million, $1.4 million, and $2.7 million, respectively [27]. The land was donated to the US in 1888 with the mandate that it would be used to establish the “Pacific Branch of the National Home for Disabled Volunteer Soldiers” [28]. Congress disbanded the National Home system and repurposed the property as hospitals and care facilities in 1958 [29]. The largest Veteran housing development in the country was finally closed in 1972 with the eviction of 2800 Veterans from WLAVA [30]. In the ensuing decades, the WLAVA leased land to interests unrelated to Veteran services, prompting a group of unhoused Veterans and the ACLU to file a 2011 lawsuit against the VA. A 2014 settlement led to the creation of the Greater Los Angeles VA Campus Draft Master Plan (DMP), ostensibly returning the WLAVA to its original goal of housing Veterans. However, the VA itself was not reauthorized to offer resources beyond medical care. Instead, the VA was empowered to subcontract with private developers to build and maintain supportive housing on campus [28]. The DMP called for 1,200 units of permanent supportive housing to be built on the WLAVA campus in a phased development plan. While over 700 new units of housing were projected to be completed by 2021, only ca. 50 new units were completed. In the meantime, Veterans Row and the VA’s creation of CTRS attempted to address the vulnerability to COVID-19 created by the housing gap.

## 2. Materials and Methods

### 2.1. Setting

Data collection was part of a quality improvement (QI) project to iteratively improve care experiences for Veterans residing in CTRS [7]. Between April 2020 and October 2021, 381 unique Veterans were admitted to CTRS, with 110 (28.9%) admitted more than once. While these numbers represent only CTRS residents and not Veterans Row, there was substantial movement and overlap between these populations, as many Veterans resided in both encampments. The average length of stay was 35 days, with a range of 1 – 365 days. The VA estimated that 20–30% of the Veterans at CTRS were amongst the top 1% of Veterans at risk of hospital admission or death [31]. Nearly all residents had a chart history of some mental health diagnosis (89%), with the majority having a serious mental illness diagnosis (SMI, 59%). Over 80% had a chart diagnosis of substance use disorder (SUD), and nearly 52% were dually diagnosed with SMI and SUD. Overall, there was substantial morbidity in this population that increased susceptibility to COVID-19 and disproportionately adverse outcomes with COVID-19. During this research period, there were only 2 documented cases of COVID-19 amongst the 381 Veterans who came through CTRS, a population that was tested frequently. There were no documented cases in Veterans Row, though there was no ongoing surveillance testing in that encampment. These low case rates suggest the encampment successfully prevented COVID-19 transmission, though a rigorous analysis of this question is beyond the scope of the current paper. 

### 2.2. Design

Ethnographic methods served as the foundation for developing QI recommendations and interventions that rapidly responded to Veteran and staff concerns. Two PhD-level medical anthropologists conducted ethnographic fieldwork in the encampments [32]. Ethnography is an anthropological study design and method emphasizing long-term immersion within a field site through the practice of participant observation [33]. Participant observation entailed regular observations, involvement in daily routines, and the development of ongoing relations with participants in the two encampments [34]. Observations, key quotations, targeted interviews, and conversations were documented by writing contemporaneous fieldnotes in notebooks [35,36]. These notes served as the basis for an end-of-the-day joint fieldnote that created consensus across the fieldwork descriptions between the anthropologists [35]. Joint fieldnotes allowed for discussing interpretations and identifying themes for quick dissemination to the larger QI team as well as follow-up questions during future fieldwork and interviews.

### 2.3. QI Feedback & Dissemination

To disseminate field reports quickly, the QI team facilitated a real-time feedback loop among multiple stakeholders, including Veterans, CTRS management and staff (hereafter referred to collectively as “CTRS personnel”), VA leadership, VA primary and mental healthcare providers, VA homelessness and health services researchers, and LAC street medicine experts. Updates and recommendations were presented during weekly QI meetings with CTRS personnel, monthly meetings with researchers, and meetings with street medicine experts. Preliminary results were iteratively discussed with participants to ensure accurate interpretations and conclusions were being drawn. 

### 2.4. Data Sources

Veteran participants in digitally recorded semistructured interviews were recruited via purposive sampling with attention paid to diversity in race, ethnicity, age, and gender as well as in diagnostic and health profiles, thus including participants with and without serious mental illness (SMI), substance abuse, physical disability, and chronic health needs. All CTRS personnel were invited for interviews. The demographics and diagnoses of CTRS participants were characterized using data from the VA’s Corporate Data Warehouse and homeless registry. 

### 2.5. Data Collection 

Using physical distancing and masks, 65+ onsite participant-observation visits were made at least weekly from 1 September 2020, to 1 October 2021, with episodic visits into December 2021, yielding 450+ pages of fieldnotes. These data were integrated with semistructured interviews (60–90 min each) with 21 unique Veterans who resided in CTRS and/or Veterans Row and follow-up interviews with 7 Veterans for a total of 28 Veteran interviews. Semistructured interviews were also conducted with 11 CTRS personnel with 6 additional follow-up interviews for a total of 17 CTRS personnel interviews (60 min each). Rapid ad hoc targeted interviews were conducted to answer specific questions posed by the larger QI team. Targeted interviews were transcribed in our fieldnotes verbatim in real-time and/or from memory as quickly as possible. 

After two months of fieldwork, the QI team drafted interview questions that explored participant issues and concerns. Veteran interview guide domains included: (1) background and homeless history; (2) previous experiences with VA clinical and housing services; (3) experiences of integration and/or exclusion in CTRS and/or Veterans Row; (4) satisfaction with facilities, care, and services at CTRS, Veterans Row, VA, and outside the VA; (5) expectations for care and housing. While interviews ideally followed a semistructured format, they deviated from this structure at times for reasons including intoxication and psychosis. An open-ended, person-centered approach was used to explore these experiences as they contributed to our understanding of what it meant to be an unhoused Veteran at this place and time [37]. The CTRS personnel interview guide domains included: (1) background and experience in developing homeless initiatives; (2) goals and purpose of CTRS; (3) benefits and challenges of providing services in an encampment; (4) experiences serving Veterans in an encampment; (5) recommendations for improving care and services. 

### 2.6. Data Analysis

Grounded theory informed data collection and analysis throughout the project [38]. As a theoretical framework, grounded theory iteratively tests questions and hypotheses over time. This approach accounted for the changing requests and recommendations of Veterans and CTRS staff as QI interventions were refined. Data analysis included the process of triangulation through collecting several kinds of data from the same sources over time, including semistructured and brief targeted interviews, repeat observations of field sites as well as from independent sources such as media, VA reports, the EHR, and VA providers to verify the validity of findings. 

Recorded interviews were professionally transcribed. Fieldnotes and interview transcripts were summarized and integrated using rapid turn-around analysis. Feedback and cross-checking with the QI team, Veterans, and CTRS staff during regular presentations of data contributed to developing a codebook derived from the interview guides [39]. Interview and fieldnote coding were organized using ATLAS.ti software. Content analysis principles were used to characterize Veteran and VA personnel reflections and experiences [40,41]. 

## 3. Results

### 3.1. Veterans Row: A Safety Net

Since at least 2018, unhoused Veterans had been sleeping on the street just outside the western gates of the WLAVA, the site that would become Veterans Row. A former resident from Veterans Row described discovering the site two years before the pandemic. He had traveled to the WLAVA for residential treatment services. When the program would not accommodate his service dog, he spent the night on Veterans Row:

I see all these people lined up and tents and people sleeping on the sidewalk. And so, I just walk over to see what this is and its Veterans! They say, “What’s up, man, what happened to you?” And they’re laughing about it. They ask, “They’re not letting you in?’” And I say, “No, they told me to come back tomorrow.” They say, “Well, we got you, this happens all the time.” I mean it was common knowledge and I was blown away. I thought, “What’s going on?”…They gave me blankets, they gave me something to eat. They told me where everything was at.

What became known as “Veterans Row” during the pandemic had been a place for Veterans to sleep and care for one another for years. Veterans Row became an informal safety net for those seeking VA services or who had been discharged from the ED, hospital, or residential programs. Veterans on Veterans Row felt abandoned by the VA, but Veterans Row itself was not a space of abandonment. It was a place for Veterans to find a modicum of care and community.

Veterans Row was a site of intergenerational kinship and camaraderie. One Veteran discovered his aunt living there while he was enrolled as a service user in a residential treatment program on campus. He recalled, “When I was in the [The Domiciliary Care for Homeless Veterans Program] in 2018, nine tents were there. I found my auntie out there.” She was dealing with bipolar disorder and alcoholism and had just left her husband. Veterans on Veterans Row helped her enter a VA program. Veterans Row reminded him that Veterans like him are chronically precarious, “one bad day away from a tent.” 

Prior to the pandemic, Veterans Row was periodically forcibly removed by the LAC Sherriff’s Department with bulldozers, often prompted by complaints from the local community. In response to a December 2019 sweep, these Veterans returned to the courtroom, and in February 2020, they delivered a sworn declaration regarding the mistreatment of unhoused Veterans. They had been asking for permission to camp on the VA property in 2019. Sympathetic VA administrators explored the possibility of temporary encampments or mobile housing on campus with wrap-around supportive services, but it took the sudden emergence of the COVID-19 pandemic to prompt implementation of such a plan. Simultaneously, the LAC moratorium on encampment sweeps provided Veterans Row with a beachhead to organize, network, and ultimately expand as both a protest of the VA denying onsite housing and an informal demonstration project for a Veteran-driven model of care.

### 3.2. Pandemic Response: Repairing Relationships

Tentative explorations for Veteran encampments became a blueprint for action as the WLAVA received emergency authorization to allow Veterans to camp on campus, with the justification that outdoor congregate living did not present the same COVID-19 transmission risks as indoor living and that a temporary outdoor tent shelter did not count as housing. A small parking lot area, unused thanks to the rapid transition to telecommuting by VA employees, was designated as a space where up to 25 Veterans could camp in their own tents. The emergency initiative was conceptualized as a medical intervention and paid for by health systems funding. Therefore, all nonmedical services, such as food, had to be provided by outside entities. Community groups, VA Voluntary Services, a local private school, and the WLAVA’s academic affiliate offered support, and the encampment soon provided 3 daily meals, tents, sleeping bags, and clothing. The VA provided limited healthcare, case management services, and contracted security. 

CTRS began as a “low barrier” to entry initiative, with no curfew or sobriety requirements in particular, a novelty amongst local transitional housing and shelter programs that often discharge residents for repeated episodes of behavioral disturbance, intoxication, or alcohol or drug possession. CTRS personnel understood that while implementing pandemic safety precautions was paramount (e.g., handwashing stations, bathrooms, laundry and shower services, masks, social distancing, and eventually surveillance testing, etc.), another critical purpose of CTRS was to repair relationships with Veterans as a prerequisite to connecting them to healthcare and housing services. One staff member explained: “Develop relationships, have the care that people need, whether it’s substance abuse, mental health, primary care. Help people navigate and get connected to those [services] they have a right to. Let’s repair your relationship with the VA…It’s the foundation of good healthcare.” The process of developing trusting relationships entailed, in the words of another staff member: “[Giving Veterans] the time and the space and the support to figure out what it is that they want.” 

### 3.3. Veterans Pushing the System

In the early days of the pandemic, CTRS became an attractive destination because of its low-barrier approach, which included regular access to necessities. Some of CTRS’ growth was due to word-of-mouth and Veteran self-referral. VA programs also referred unhoused Veterans to CTRS. Relatedly, pandemic conditions impacted the capacity at other VA and LAC shelters and housing programs, options that often shut down or had limited capacity. 

While VA staff outreached to Veterans Row daily, inviting residents to the inside encampment and offering them other housing options, most Veterans did not accept these invitations. As the CTRS census grew, the Veterans Row census grew as well. Some Veterans chafed against the rules at CTRS, such as the mandatory security search upon each entry to the encampment, a prohibition on guests, limited accumulation of personal belongings, and weekly COVID tests. Others complained that tents on a concrete parking lot were too small, uncomfortable, and undignified, requiring elderly and disabled Veterans to bend over to enter them. One Veteran dismissed CTRS as “tiny little shit pup tents.” 

Veterans Row accepted donations of larger 10’x14’ tents. The tents were spacious and looked tidy, uniform, and organized next to each other, with an American flag affixed to the front of each tent. Photographs circulated in media reports of what some characterized as defiant Veterans fighting a righteous cause, and this imagery staved off rumors that disparaged those sleeping on the street. A Veteran explained why the appearance of large, orderly tents with the American flags was important:

We put flags on them to show that these were Veterans because there was…a lot of this public ignorance being built [by local residents] And they’re posting on social media…[that] they’re not Veterans, they don’t have benefits, they’re criminals. And we’re like, okay, we got the flags on them. No, these are Veterans.

The larger tents took on multiple purposes in the early days of the VA. They provided Veterans a comfortable place to reside to help foster a better community and home. The tents also countered perceptions that these Veterans were “dishonorable” or “criminals.” They projected a sense of organization, industriousness, and tidiness from a community of Veterans deserving of the home owed to them due to their service, combating dominant narratives about homeless encampments rooted in individual blame, criminalization, and reinforcement of substance use and mental health-related stigma. Veterans Row attempted to demonstrate how to provide a home and cultivate a community while, at the same time, offering the same, or better, provisions and services as compared to CTRS.

While many deplored the sight of the encampment in the affluent neighborhood and repeatedly sought its removal, a few local residents responded with concern. Veterans on Veterans Row outreached to the Brentwood residents who pulled over their cars and asked about the situation. A Veteran explained, “You need to stop and talk to people. So we do that [with] people in the community.” Interactions with residents raised awareness that Veterans Row residents wanted the VA to provide homeless Veterans with a home on campus. Veterans also solicited volunteers, donations, and further networking opportunities. Donations of a 40-foot-long dumpster, portable toilets, and handwashing stations with regular service and maintenance helped keep the sidewalk and street clean. Veterans Row did not want their “dirty” status used as a reason to ask the LAC Sheriff to bulldoze their encampment despite the moratorium. A Veteran said:

If I leave [garbage] in the street, they’re gonna come through and plow this shit out. That’s what they’re gonna say because this is always their excuse, “It looks like shit, get rid of it. There’s feces. There’s this, there’s that.” So I’m trying to stop that from happening, okay?

Veterans Row was continually improving through the involvement of the Veteran residents as well as outside benefactors. Veterans Row had recognized leaders and a loose leadership council. These leaders facilitated partnerships with local restaurants, bakeries, and caterers to deliver daily meals. They also put on a successful fundraiser that raised $4000—an exhibit of photographs of Veterans Row residents and art made by residents. Veterans Row leaders created a census and tracking sheet of Veterans and their partners/family members residing in the 45+ tents. They developed intake and orientation procedures for newly arrived Veterans. Non-Veterans were not welcome unless they were Veteran partners or family members. A list of duties and tasks were assigned to the residents: fire watch, dumpster detail, street clean-up, flag detail, and visitor and donation assistance. They created their own incident report forms. A military sensibility was palpable in these designations, roles, and hierarchies, especially in an expressed vigilance for the infiltration of “insurgents” intent on sabotaging Veterans Row. One resident shared that he had experience working undercover, and he expected VA spies had already infiltrated Veterans Row. Order and community were actively monitored and enforced. 

Veteran advocacy organizations supported Veterans Row, as did a few former residents who had obtained permanent housing. One former resident explained how they developed relationships with a range of agencies within and outside of the VA, connecting residents and often recruiting them to services related to housing, substance use, medical, and mental health needs. They facilitated housing placements for Veterans but also found that many of these Veterans returned because “people want to be by the VA.” Volunteers also spearheaded collective recycling efforts for income generation. Veterans on Veterans Row pointed out that their community worked because they had ownership of it. In the words of one Veteran, “we take care of each other and ourselves,” while in many programs, including CTRS, they were treated “like children.” 

While Veterans in Veterans Row worked to maintain a tidy, hygienic area, they emphasized that it was still a homeless encampment unsuitable for human habitation. They leveraged a dense web of meanings through their bodies and residence. They made visible their unimpeachable status as Veterans, in many cases medal-awarded combat Veterans, embodying American ideals of patriotism, by welcoming media coverage from all political slants, from local news affiliates to the far-right Epoch Times and Fox News, to NPR, to “citizen-journalists” posting images and commentary on social media. They also made visible the dirt and wounds on their bodies. A notable example was a male Veteran, wearing a tank top, displaying a 5-inch self-inflicted gash in his upper arm, a deep wound in need of sutures, visibly infected. Despite clearly experiencing pain, he lifted heavy items, swept, and cleaned outside the tents in view of all – outreach workers, reporters, VA employees, and cars that sped by. Shaking off offers of care, he simultaneously displayed his vulnerability and his industriousness. 

Veterans were aware of the paradox of creating an organized encampment while also rendering visible to the public how residing on the sidewalk is unsafe, unsanitary, and, ultimately, an injustice worthy of protest. They posted signs reading “no pee zone” and chastised a resident who dumped his urine into the gutter outside his tent, creating smelly, unsightly streams. The Veteran protested that he did this because his physical disability rendered him unable to walk to the portable toilets at the end of the row. Bathing was a challenge for residents, as another explained: “I’m bathing in a bucket in a tent.” Washing his body, he told us, was a way to wash away negative thoughts: “personal hygiene does a lot for mental health and a lot for keeping up the drive. Any drive I have to be productive…just gets washed away because I can’t wash away. You know?” The dirt accruing on his body impacted him deeply. The difficulty accessing shower facilities structured the very existential possibilities he felt his life had. A clean body helped him “keep up the drive” to live. This drive to live was implicated in a suicide attempt prior to his arrival on Veterans Row: “if my mental health isn’t there because of something simple like I can’t wash, how much more am I going to do for myself if I can’t even do that?” He emphasized the impact of not having access to basic hygiene resources as it eroded his sense of self and his desire to “be productive” to participate in mainstream American society as a worker and a consumer. Veterans Row residents wanted observers from local communities and the media to understand the daily challenges they faced and to help them advocate and amplify the narrative that the very dirt and wounds on their skin, the explosion of substance use, which many told us was a way to calm their nerves from chronic stress or to delay sleep to avoid nightmares and the vulnerability of sleeping on the street, and the many dangers they were exposed to were the result of not having the safe home they were entitled to. As one Veteran put it, “What do you want me to do?…But this situation, you guys are *seeing this*, right?” 

Even though Veterans Row created a community through advocacy, political action, and networking, residents were still supportive of CTRS. One Veterans Row leader declared the creation of CTRS a momentous achievement: “I think that CTRS is awesome—it was like taking the Beaches of Normandy, getting the guys back on the property.” He encouraged Veterans to use CTRS: “I’m always trying to sell Veterans on going inside.” The hope for those residing at CTRS was that the VA would eventually upgrade them to a better home. In this way, Veterans on Veterans Row were also waiting to be welcomed home and hoped to be able to one day abandon their encampment, but on their terms. They were not simply antagonists but optimists that they could push the VA to do the right thing and make Veterans Row obsolete by providing an on-campus community—a home—for unhoused Veterans.

### 3.4. VA Responds: Building a Veteran-centered Community

In response to positive early experiences, growing demands from Veterans to improve the CTRS environment, and the increasing impacts of an ever-growing pandemic, the VA moved the encampment to “The Great Lawn,” a large field near a shaded rose garden with tables and benches. There was more space in this setting, and for staff, the move gave the initiative a sense of permanence. Large tents similar in size to those on Veterans Rows replaced the small tents, and they were placed on raised wooden platforms to keep them off the wet grass. Each tent had a cot, bedding, storage locker, clothes, and a range of personal items available by request through VA Voluntary Services. Gravel pathways laid down between rows of tents allowed for easier movement. Floodlights and security cameras were installed, and a 24/7 contracted security detail was positioned at the entrance and around the field. Additional tents were erected in a paved parking lot at the entrance to CTRS for Veterans with limited mobility. A laundry service was provided for several months. Veterans were transported daily to showers at either the VA Welcome Center or YMCA by CTRS staff until an onsite shower trailer began service in January 2021. Veterans could receive mail, a vital service that facilitated the completion of the extensive paperwork applying for housing and other services entailed. Onsite urgent care services were provided every other week, and staff attempted to facilitate Veterans making their healthcare appointments. An onsite physician provided COVID tests to new admissions. Encampment capacity expanded to 100, though its largest census as a tent encampment was ~60.

The provider area—several tables on a raised platform under an open-air canopy—was initially in the center of the encampment, along with a coffee and snack station and a phone charging area. While CTRS personnel reported some difficulty working in an open-air environment during hot summers and blustery winters, this area became a place for Veterans and personnel to socialize. One staff proudly pointed out that the lack of doors and walls facilitated a spirit of transparency, access, and collaboration with Veterans. Veterans gathered near the open-air provider tent in the morning to make coffee and linger in conversation. Many Veterans ate together and started their day in this communal area. Personnel reminded them about upcoming appointments regarding healthcare or housing. One Veteran regularly shared anecdotes about riding motorcycles. Another insisted on receiving his “morning hug” from a Veteran peer specialist. Veterans also placed locally scavenged couches and tables at the front of the encampment under the shade of tall pine trees. This “lounge” area facilitated Veteran socializing.

### 3.5. From Community to Dumping Ground

A local spike in COVID cases in December 2020 led to fear of an outbreak, a period of limiting new admissions, increased enforcement of masks and social distancing, and testing within CTRS. Most of the original staff had turned over by this time, the 10th month of operation, and the early enthusiasm and idealism about the potential for CTRS were fading. Personnel expressed that there was a lack of consensus around the mission of CTRS from VA leadership and felt unsupported and understaffed [7]. There was growing exhaustion about the regular interpersonal conflicts among Veterans and between Veterans and staff, mental health crises, VAPD involvement in disputes and psychiatric holds, and the day-to-day labor of setting up and taking down tents and cleaning hazardous biomaterials such as animal and human excrement. These factors contributed to burnout and low morale. Staff responded by limiting admissions and increasingly discharging Veterans for rule violations as the census dipped into the twenties. During one encounter, CTRS denied admission to a Veteran with SMI who had refused a nonmedical community-based residential placement (a “board and care”) and was exhibiting “disorganized behavior.” Personnel felt they could not accept Veterans like these without clear housing plans, given that they “need more help than we can provide” and that there was no onsite mental health staff. Personnel concluded that the referral was “a frigging dump,” expressing a common theme at that time amongst personnel that CTRS had become a “dumping ground” for other VA programs to send undesirable patients that no one wanted to be “stuck with.” Personnel then explained how Veterans exhibiting signs of mental illness would lock themselves in a tent and would not communicate with staff. These patients, they said, never “tried any housing options” and were “too complex to be here.”

By January 2021, Veterans were discouraged from socializing at the provider tent. Instead, Veteran discussions with the staff needed to focus on individualized “housing plans.” The “lounge” was removed over concerns that socializing encouraged illicit drug use. This left Veterans few places to socialize in CTRS. A Veteran said that it was “dead” and lacked the community spirit that once existed. Another Veteran told us how focusing on the “other stuff” such as housing plans rather than Veteran morale was a “waste of time”:

You can look at their faces—our faces…and you don’t see a lot of smiling faces. So that’s one thing that could really use some help and improvement is building morale and trying to figure out why there’s not many smiling faces…because all the other stuff would be probably a waste of time if everybody’s still angry and upset.

CTRS personnel worried that “getting too comfortable is a problem here. [This Veteran] has all this stuff in [his tent]. When he leaves, we’ll have to clean up all that stuff…We are cleaning urine and feces and we are tired of it. It’s a problem.” Personnel shifted the initiative’s focus to quickly transitioning Veterans to alternative housing (e.g., motels, programs, or apartments). They were concerned that if too many services were provided, Veterans would feel encouraged to permanently reside within the encampment. They said, “I’m sympathetic to Veterans…and the traumas they face [but] we’re only 50% of it. They’ve got to meet us halfway…” Another bluntly said: “You’re never supposed to work harder than the Veteran trying to get his recovery. The Veteran has to put some blood into it.” For some personnel, comfort and community were associated with substance use, as indicated by concerns that “none of these people will seek services on their own if they are just allowed to be here, getting high” and “a lot of these guys aren’t going to go nowhere until they’re forced.” They advocated brief CTRS stays as a transition to more permanent housing programs. Discharges for rule violations were described as incentives for Veterans to practice more self-control and responsibility in order to transition into better housing.

Many Veterans left CTRS for Veterans Row by January 2021, either voluntarily or through involuntary discharge. Some Veterans “double tented”: kept a tent in each encampment to access the benefits of each site. CTRS began forcing residents to choose one encampment or the other around December 2020. A discharged CTRS resident residing on Veterans Row situated the failure to care for unhoused Veterans like him within systemic societal failures: 

The problem is that we fail time and time again just as being derelict in our own duty to the person in front of us, derelict to the idea that every person that is sitting in a bed that is a Veteran that needs help, receive help and who has a voice should be heard. And rather than being heard are being muted, are being left out, are being discarded, are being cast away, are being sent in the opposite direction, are being sent in a worse system of existence. 

Many Veterans felt CTRS had become too restrictive and that it no longer practiced the low-barrier, harm-reduction approach it originally aspired to. Veterans Row welcomed all Veterans, but the community had its own rules. One resident was asked to leave after he was caught stealing from other residents. Another was kicked out for leaving too many of his collected recyclables strewn around the encampment. Already on thin ice because he refused to join in collective recycling efforts, his untidiness seemed to be the last straw. Veterans Row was potentially combustible: a community of combat-trained Veterans with high rates of substance use, trauma, chronic distress, and poverty, living in close quarters under the scrutiny of the VA, the sheriff’s office, the media, and the local community. Conflicts were usually settled without resorting to blows or calls to the authorities. 

### 3.6. Without Home: Social Abandonment

VA staff hoped to convince all the Veterans Row residents to move into CTRS. They assumed that those who refused to move inside preferred the easier access to illicit substances they had outside the gates. Substance use was a part of the social ecology on Veterans Row. However, Veterans expressed that what they valued most was both the camaraderie and their autonomy, including involvement in various economies (e.g., recycling, bicycle maintenance, art, music, and drugs). One Veteran’s tent was full of computers and electronics, and another was a musician. Many Veterans felt safer with weapons hidden in their tents, which would have been a felony on the VA campus. One Veteran whose disability impacted his ability to walk and led to his discharge from CTRS repeatedly returned to Veterans Row after stints in the hospital and skilled nursing facilities (SNF), which was against the advice of some of his fellow Veterans. He explained that he returned because he felt controlled and disrespected in institutional settings. He felt his presence was valued on Veterans Row, where he had been since prior to the pandemic and where he was proud to be referred to as “one of the founders.” 

Several Veterans found it difficult to leave the communities of their respective encampments. Despite obtaining a permanent apartment, one Veteran explained that he stayed in CTRS because he had a series of procedures and appointments upcoming at the VA, and it was more convenient to stay on campus. However, he also had developed relationships with other Veterans at CTRS and appeared to seek out their company. One Veterans Row resident suffered a panic attack on his way to tour a potential apartment for rent and demanded he be returned to his tent as he was not ready to leave. Another compared the solidarity he experienced in Veterans Row and the duty he felt towards his fellow unhoused Veterans to the solidarity he felt during deployment in Iraq. Two members of his unit went missing, and despite exhaustive searches, their bodies, buried in unmarked graves, were not recovered until a year after he had returned home. He explained he carried that experience to Veterans Row: “I didn’t like the idea of leaving like that. And then with this situation…I don’t like the idea of leaving anyone behind.” He continued to advocate for the residents on Veterans Row and to be an active part of the community, even after obtaining housing. He did not want to leave any Veteran behind on the street. 

Indeed, “not leaving any Veteran behind” took shape in concrete ways. Many Veterans Row residents carried intranasal naloxone, and a number had CPR or military medic training. The authors witnessed a community response to an overdose one morning when they responded to calls for help from a tent. A Veteran was carried out, pulseless and cyanotic. They called 911 and initiated CPR. Many Veterans joined the response, delivering CPR and administering two doses of intranasal naloxone, reviving the Veteran in a small-scale instance of a successful collaboration between VA and Veterans Row personnel. While mortality amongst the unhoused in LAC dramatically increased during the 1^st^ year of the pandemic, not due to COVID but to fentanyl overdoses [42], there were no overdose deaths on Veterans Row.

A Veteran tearfully spoke of the isolation he suffered while enrolled in a county initiative providing temporary shelter in unused hotel and motel rooms. He returned to Veterans Row after attempting suicide in the motel room where he was placed. His preference for living in an encampment community on the street, rather than staying in a motel or apartment alone, spoke to the dangers of isolation and of the important role of community for these Veterans. One Veterans Row resident killed himself in the motel he was placed in during the pandemic. Housing placements, temporary or permanent, could be in disparate locations of a vast county, leaving individuals isolated from the communities to which they felt they belonged. These placements did not acknowledge that unhoused people rely on their communities for support and companionship and that even though they desired a roof over their heads, they considered these communities their home.

### 3.7. Debating Home

As the pandemic response transitioned from crisis to routine, pressure from the surrounding community to remove Veterans Row grew, especially as the dangers of life in the encampment became more evident and led to negative publicity. Traffic sped past the encampment, and twice automobiles careened out of control into the tents, causing destruction and injury. While these incidents had no fatalities, disagreements among Veterans Row residents led to two well-publicized murders in May and September of 2021. This occurred against a backdrop of the everyday violence of involvement in underground economies that were a part of the survival strategies of life on the street, despite Veterans’ own efforts to enforce order and a communal spirit. There was no doubt that life on Veterans Row entailed considerable risk. Nevertheless, the fact that it was preferable to so many Veterans spoke to their desire for an alternative communal form of housing.

The encampment sweep moratorium ended soon after the pandemic’s 1st anniversary. Many on Veterans Row understood their days were numbered, and their efforts to find housing redoubled. They worked with VA programs as well as nonprofits and county agencies. A group of nearly a dozen residents worked to relocate to a supportive housing project together. A series of monthly public town hall meetings were initiated over the summer of 2021 by the VA administration in a parking lot between Veterans Row and CTRS. These meetings highlighted the potential for collaboration as well as fundamental discrepancies in perspectives between the VA and Veterans. 

A woman Veteran expressed frustration regarding her housing search: “I have an apartment finally, after a year and a half on a waiting list. The apartment has been ready [for 3 months] but I cannot get into the apartment because the housing authority is saying [the VA doesn’t] have the people to process the paperwork!” She contextualized her own predicament by emphasizing the structural difficulties individuals face in navigating the housing process in LAC. An administrator referred her to a staff member who was present and offered to contact the housing authority. Within a week, she moved into her apartment. This underscored both the VA’s strengths and blind spots. They could respond rapidly to individual-level service issues on a case-by-case basis but were not able to systematically address the broader obstacles individuals faced when confronting the maze of different agencies, institutions, and gatekeepers that needed to be navigated to provide Veterans housing in Los Angeles. This resulted in a bracketing out or even disregard of structural issues. 

Veterans at town hall meetings consistently brought up the contradiction that VA land was being leased to non-Veteran entities for purposes not connected to Veteran housing or health. At times, these protests were visceral and angry: “We served our country. Why doesn’t our country serve us?”; “I just want to know when the fuck I can get my land back!” and “I feel like they are stealing land that Veterans actually own.” One Veteran read the history of Veteran housing at the GLAVA from the Draft Master Plan, adding:

We’ve got 50 years of this land being neglected and it’s been used for tax write-offs, for grantee donations, and everything that’s been used has been milking the cow, taking our federal grant money. And we don’t see no housing. We need housing. This land is…to be used for housing for our Veterans and medical services, which the VA is responsible for…This is not your land. It is our land.

The administrator’s response articulated the legal constraints on the process by which the VA could provide housing, acknowledging the slow pace and the many obstacles. A Veteran interrupted, reminding everyone of the precarity of living on the street, shouting: “And a Veteran dies! A Veteran on a street dies!” This interaction reflected the slow pace of change at the VA that so frustrated Veterans and their advocates. 

After one town hall meeting, a VA representative remarked, “I think part of the problem is that they don’t focus on their recovery.” The representative questioned the preoccupation with land use, adding, “let’s get some treatment…then when you’re better, then you can [protest]…They are aggressively homeless… [and] want to stay homeless as part of this protest.” This representative felt the protest and demands around land use were distractions and that the Veterans should be focusing on their own recovery, which was an individual process involving treatment for biomedical disorders. For this representative, “aggressively homeless” meant that Veteran activism was misguided and lacked a sense of self-responsibility that impeded the process of recovery. “Aggressively homeless,” however, could also be seen as an apt description of the advocacy tactics Veterans Row deployed. They highlighted the negative aspects of their existence that were all too public and visible—the abject filth, poverty, vulnerability, substance use, and violence—as evidence of broader society’s neglect. In contrast to the VA representative, they saw political mobilization around creating a home for Veterans as critical to their recovery efforts. The Veterans often pushed against the biomedicalization of their unhoused status. One Veteran pointed out that while their homelessness was often attributed to substance use, mental illness, and individual failings, the high-paying union jobs veterans used to receive had been replaced by less secure, low-wage labor: “We can’t support our families, we get left [by our families], we have PTSD, we lose it, then start drinking and doing drugs.” 

Despite these discordant perspectives, the town hall meetings identified common ground, and Veterans and VA personnel alike expressed gratitude for the forum. Eventually, the Veterans Row encampment was demolished by the LAC Sheriff’s Department. However, the VA was able to accommodate nearly 50% of the residents in a new tiny shelter village, replacing the original CTRS tent encampment. These Veterans were granted their request to keep their community intact, as they were all housed together with indefinite lengths of stay in what was termed Veterans Row 2.0. During this time, the QI team helped improve meal quality and delivery services by leveraging a pre-existing partnership with UCLA dining services. A partnership with VA Whole Health offered community-building activities such as a healthy teaching kitchen, storytelling, gardening, and battlefield acupuncture. An “encampment medicine team” was mobilized for onsite medical, mental health, and harm reduction services (e.g., naloxone training and prescriptions, fentanyl test strips, and plans for clean needle exchanges). Following Veteran requests, an open-ended length of stay with no barriers to entering or remaining was instituted. Tents were replaced by tiny shelters (8x8 private lockable cabins with electricity, heat, A/C, bed, and desk). A Veteran-run Veteran engagement committee was created for residents to provide feedback to management. 

## 4. Discussion

Infectious disease epidemics are social processes with complex structural and environmental factors that manifest in locally specific ways [14,43,44,45]. In *Purity and Danger*, Mary Douglas famously noted that dirt is “matter out of place” [8]. During the pandemic, unhoused Veterans made visible their social status as “out of place”—dirty bodies living in spaces not meant for human habitation. They were viewed with fear and suspicion and seen as outcasts and sources of illness as if they themselves were pathogens. Their presence upset local residents. Some responded by demanding their removal, others by offering support. Veterans emphasized why they were “out of place,” shedding light on society’s inability to house and care for them despite promises made. They demonstrated their “purity,” that they were a community worthy of care, by elevating their status as Veterans and their ability and desire to care for each other.

The Veterans’ perspective of their vulnerability resonates with the syndemics framework that goes beyond risk groups and behaviors to encompass “environments of risk and agents promoting risk” [46] (p. 157). By combining the concepts of “synergy” and “epidemic,” the syndemic approach acknowledges that diseases in a population “occur neither independent of social and ecological conditions nor in isolation from other diseases” [14] (p. 8). In this view, the impact of the COVID-19 pandemic on these Veterans demonstrates how historical conditions and social relationships shape disease processes, including perceptions of contagion, through political-economic, structural, and environmental factors. By being more transmissible indoors versus outdoors, the SARS-CoV-2 virus brought the relationship between health and wealth into sharp relief and, for a time, shone a spotlight on the plight of the unhoused. 

The anthropology of infectious disease has highlighted how narrow biomedical and individualized responses to syndemic problems can exacerbate the existing social inequities that create vulnerability to epidemic infectious disease in the first place [45,47]. In this case, Veterans called for addressing the root cause of their vulnerability to COVID-19: their homelessness. They argued that the disjuncture between their idealized Veteran status and their own abject conditions pointed to their abandonment by society. In leveraging their status as Veterans, they demanded to be seen collectively as a community who had served their nation and were deserving of a home on the VA campus. 

Their encampment was not just a site of pathology and risk but a source of vitality, care, and support. PEH can and do constitute and belong to communities that provide substantial benefits even while amplifying certain harms. As Bourgois and Schonberg demonstrate in their ethnography of homeless encampments, obtaining and using drugs requires developing relationships and occurs within the context of “a community of addicted bodies that is held together by a moral economy of sharing” [48] (p. 6). Like the encampments they studied, Veterans Row was structured around the logics of exchange, hierarchy, and belonging that brought forms of care as well as harm. Well-meaning service providers focused on the harms of drug use and interpersonal violence, arguing Veterans would be better off housed away from their fellow substance users on Veterans Row. However, this approach risked pathologizing Veterans Row and made some Veterans vulnerable to overdose, isolation, and despair.

In his ethnographic study of an informal “asylum” in Southern Brazil, anthropologist João Biehl introduced the term “zone of social abandonment” to refer to a space of “social death” where “in the face of increasing economic and biomedical inequality and the breakdown of the family, human bodies are routinely separated from their normal political status and abandoned to the most extreme misfortune, death-in-life" [49] (p. 38). Marrow and Luhrmann use the term to describe the plight of unhoused individuals with SMI in the United States [50]. This article documents a case of twin encampments of unhoused Veterans that, during the COVID-19 pandemic, created forms of community and care that bolstered collective survival. For these Veterans, the zones of social abandonment were spaces of isolation and segregation, the temporary motel rooms, and the individual apartments that removed them from their Veteran community. There were multiple suicide attempts and overdoses in these placements, several leading to deaths. These sites provided walls, a locking door, and a roof, but they were not homes. Having a home for these Veterans meant being integrated into a community where they were respected and to which they contributed and belonged. Housing assignments must consider how unhoused individuals become (or fail to become) integrated into various communities and seek ways to preserve those connections [51,52]. 

Veterans Row protected against the harms of isolation, despair, and solitary drug use. However, it exposed residents to relational forms of violence, reflecting tension around space, resources, and power in the encampment. The most visible tragedies of Veterans Row were two murders perpetrated in the context of relationships and disagreements. This points to a vulnerability that would benefit from intervention but does not invalidate the beneficial aspects of the community. 

Initially, CTRS staff attempted to build a sense of community, acknowledging the power and potential of Veteran camaraderie. As CTRS developed, staff faced challenges with understaffing, turnover, and burnout, exacerbated by the difficulties caring for medically complex Veterans. CTRS personnel worried about Veterans becoming “too comfortable” and began transitioning away from the initiative’s initial low-barrier model. During these times, many Veterans left CTRS, preferring the communal encampment outside the gates. 

However, with the LAC Sheriff’s Department signaling their intent to demolish Veterans Row during the summer of 2021, VA administrators and CTRS management diligently and empathically worked together to re-envision CTRS as a hospitable place for these Veterans to stay. CTRS personnel and the QI team collaborated with Veterans to create a renewed sense of community. VA administrators acted on Veteran feedback through the ethnographic research of this QI project. 

The result was not a perfect, conflict-free encampment but an ongoing effort by the VA and Veterans to preserve a space of dialogue, discussion, and experimentation, attentive to safety, risk, relationship, and community. While the dialectic of the dual encampments ended with the encampments ultimately being collapsed into one, the process of debate and negotiation between Veterans and the VA continues. Encampments can provide residents benefits, including safety, autonomy, and stability, as well as an opportunity for health and housing services to engage with vulnerable communities [53,54]. It is also important to recognize that there is no single approach that fits everyone, as different approaches suit individuals differently depending on the circumstances [48].

While CTRS is a bold attempt to respond to the ongoing crisis of Veteran homelessness during a pandemic, it also is a reminder of the ongoing lack of adequate housing for Veterans. The Veterans’ frustration with the slow pace of housing construction has resulted in a new lawsuit against the VA [55]. CTRS is a potential model for other medical centers and communities across the country to rapidly respond to the syndemics of homelessness and COVID-19 and for the many other comorbidities, medical and social, faced by the unhoused. The model works best when a sense of community is fostered and when encampment residents play a significant role in articulating the project’s mission. 

Unhoused Veterans were pathologized, often treated as pathogens themselves. The Veterans simultaneously fought against and instrumentalized their identity as pathogens to make demands for care. They contested the notion that they were pathogens when they evoked their status as Veterans, patriots who sacrificed themselves to defend the “American way of life” but were abandoned by their country. They demanded that the VA fulfill its promise of giving them a home. Moreover, they also leveraged their reputation as pathogens, feared and despised in one of the most affluent zip codes in the US, as a means of foregrounding the injustice they suffered. These Veterans are not just patients in a healthcare system but political actors in ways that resonate with other social movements for recognition, care, and social justice, such as the activism around HIV/AIDS treatment and the Occupy Wall Street Movement [56,57,58]. They were conscious of their privileged status as Veterans and did not seek to tear down the VA. They wanted the VA to live up to the highest ideals of its mission, as stated in the Draft Master Plan to “restore and improve the site to play the role for which it was historically established” by providing them with an on-campus home [28].

## Data Availability

Data is unavailable due to privacy and ethical restrictions.

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
