# Peer review of "Purity, Danger, and Patriotism: The Struggle for a Veteran Home during the COVID-19 Pandemic"

_pathogens, 2023, doi:10.3390/pathogens12030482_

Round 1
Reviewer 1 Report
The paper analyses the modes of relationship and the production of normativities in two homeless encampments in the context of the COVID-19 pandemic. Through an ethnographic approach, the paper describes the transformation of the normative meanings and identifications of a wide range of agents, highlighting how these transformations construct the category "home" among camp residents, institutional workers, and political representatives.
This paper addresses a relevant issue and contributes to a topic that has been little explored in the ethnographic literature. The paper also stands out for its ethnographic emphasis in a field in which access to informants and research itself poses considerable difficulties.
The paper is clear and well presented. The methodology conforms to ethnographic rigour, and the results are relevant.
As a comment to the authors we suggest to expand the literature on which the Introduction is grounded:
1. The concept of "homelessness as social pathogen" (line 51) and Mary Douglas' notes on "dirt as matter out of place" (line 675) could be brought into dialogue.
2. The conceptualisation of "home" (lines 93 to 116) could be broaden. Thus, the contextual and situated dimension that is developed throughout the article could be explored in greater depth already in this section of the paper. We encourage the authors to (briefly) present the references on which they base the discussion of their results.
Second, we consider that the categories "patriotism," "ingenuity," and "vulnerability" (line 82) — as categories constructed in the field — should be enclosed in quotation marks.
Third, there are some small typographical errors:
- Add a space: "transmission[4]" (line 35).
- "3.5 Without Home: Social Abandonment" (line 531) should be 3.6.
- "3.6 Debating Home" (line 584) should be 3.7.
- Add a period: "(exchanges) Following" (line 666).
Author Response
Thank you for the generous compliments and feedback on our article. We have responded to your suggestions below.
Point 1. The concept of "homelessness as social pathogen" (line 51) and Mary Douglas' notes on "dirt as matter out of place" (line 675) could be brought into dialogue.
Response 1. We appreciate the encouragement to bring Douglas' ideas of "dirt" as "matter out of place" into dialogue with the concept of social pathogen in the introduction, and we have done this, linking the concepts and indicating how the social pathogen concept does emerge out of Douglas' theory.
Point 2. The conceptualisation of "home" (lines 93 to 116) could be broaden. Thus, the contextual and situated dimension that is developed throughout the article could be explored in greater depth already in this section of the paper. We encourage the authors to (briefly) present the references on which they base the discussion of their results.
Response 2. We broadened the literatures on the anthropology/social configurations of home and added 2 sentences to the section. The first addresses the strategies and practices of creating homes in temporary situations among displaced people and the other about the idea that homeless encampments have meaningful histories and heritage to them.
Point 3. We consider that the categories "patriotism," "ingenuity," and "vulnerability" (line 82) — as categories constructed in the field — should be enclosed in quotation marks.
Response 3: We enclosed these terms in quotation marks.
Point 4: There are some small typographical errors:
- Add a space: "transmission[4]" (line 35).
- "3.5 Without Home: Social Abandonment" (line 531) should be 3.6.
- "3.6 Debating Home" (line 584) should be 3.7.
- Add a period: "(exchanges) Following" (line 666).
Response 4: we addressed these errors, thank you.
Reviewer 2 Report
This well-written and conceptually clear paper is not really a study of the ethnography of infectious disease, but rather of homelessness and struggle for home and community among veterans. Good paper, but this special issue of Pathogens is the wrong venue. While the paper seeks to employ syndemics theory, it does not actually describe a bio-bio syndemic interaction among two or more disease. While the veterans suffered from various diseases, disease interaction, the heart of syndemics theory is not described. The paper would also benefit by discussing whether the encampments were able prevent the spread of COVID.
Author Response
Thank you for the detailed and thoughtful feedback. We respond to each of this reviewer's main points in turn:
Point 1. This well-written and conceptually clear paper is not really a study of the ethnography of infectious disease, but rather of homelessness and struggle for home and community among veterans. Good paper, but this special issue of Pathogens is the wrong venue.
Response 1. We do feel the special issue is a good venue for this paper. We show how the COVID-19 pandemic has brought to light a dense network of socio-historical inequities in new ways. This is an important aspect of the social impact of epidemics, and is not unique to COVID-19. Furthermore, it is an aspect of epidemic infectious disease that ethnography is well-positioned to highlight. Our work follows in a long tradition of scholarship in the anthropology of infectious disease that highlights the broad social contexts and dynamics that make communities and individuals more or less vulnerable to epidemic illnesses, thus we feel the special issue is an ideal fit for it.
Point 2. While the paper seeks to employ syndemics theory, it does not actually describe a bio-bio syndemic interaction among two or more disease. While the veterans suffered from various diseases, disease interaction, the heart of syndemics theory is not described.
Response 2. It is true we do not focus on syndemic interaction between 2 diseases. As the reviewer points out, however, the Veterans in our fieldsite suffered from a number of diseases that heightened their risk for COVID-19 infection and for bad outcomes due to COVID-19 and we do cite the literature that indicates this as well as the general indicators of morbidity in the Veterans who came through CTRS. We have further articulated this in the revisions and added more citations. We draw from work on syndemics and COVID-19 by Singer and Rylko-Bauer (2021) and others. For example, we write that “By combining the concepts of “synergy” and “epidemic,” the syndemic approach acknowledges that diseases in a population “occur neither independent of social and ecological conditions, nor in isolation from other diseases” [12] (p. 8). The impact of the COVID-19 pandemic on these Veterans demonstrates how historical conditions and social relationships shape disease processes, including perceptions of contagion, through political-economic, structural, and environmental factors. By being more transmissible indoors versus outdoors, the SARS-CoV-2 virus brought the relationship between health and wealth into sharp relief and for a time, shone a spotlight on the plight of the unhoused.” There is an ongoing debate regarding whether the COVID-19 pandemic can be considered to be a syndemic, and if so, where and in what ways. We have cited some of this literature, and in the spirit of these scholars, we argue that, even though we don't focus on COVID-19 and just one other disease, the 3 components of the syndemic framework do apply to our case:
Component 1: Political-economic forces with historical depth lead to entrenched social, economic and power inequities. Such forces are clearly implicated in the case of unhoused Veterans and we do sketch that out.
Component 2: Those inequities shape the distribution of risks and resources for health, leading to the concentration of disease in specific parts of a population. This is also the case for unhoused Veterans
Component 3: Some overlapping diseases make one another worse because of biological interactions. COVID-19 has worse outcomes amongst the unhoused due to the presence of multiple comorbidities directly linked to being unhoused. The reviewer points out that this component is the weakest in our paper. It is also the weakest generally in the literature on COVID-19 and syndemics. As Courtin & Vineis 2021 (full cite in revised manuscript) point out, more attention has been paid to COVID-19 as syndemic in terms of social inequalities and co-occurrence with other diseases, and less attention has been paid to the underlying biological mechanisms of the syndemicity. Nonetheless, along with the scholars we’ve cited I our revised manuscript, we do feel the three components of the concept of syndemic have a clear bearing on understanding the dynamics of COVID-19 amongst unhoused Veterans as well as on practical preventive action.
Component 3. The paper would also benefit by discussing whether the encampments were able prevent the spread of COVID.
Response 3. We did add this to the revised manuscript. We note that amongst the 381 Veterans who came through the officially sanctioned encampment, CTRS, there were only 2 COVID-19 cases. These were detected in the context of routine surveillance. There were no documented cases in Veterans Row, the "unsanctioned" encampment, but we do not have the complete numbers of Veterans who went through there and there was also no surveillance testing there since that encampment was not operated by the VA. While this does not definitively prove that the encampment prevented COVID-19 transmission, the analysis required for that is beyond the scope of our manuscript.
Reviewer 3 Report
Clear presentation of a complex issue, weaving together a coherent story that is grounded in theory and articulates the value of ethnographic approaches in understandings of infectious diseases.
Minor typos to address throughout, including:
line 533: access 'to' illicit
line 572: suffered as 'a' patriot
line 574: 'HIs' should be "His"
line 666: need a period after 'exchanges)'
line 675: missing 'of' in "matter out 'of' place"
line 766: period outside the brackets [ ]
Author Response
Thank you for these generous remarks. We have addressed each of these minor typos.
Round 2
Reviewer 2 Report
The authors have successfully responded to earlier criticisms and revised the paper sufficiently to warrant publication.
Reviewer 3 Report
As this paper isn't focused on syndemics, rather the contributions of ethnography to studies of infectious diseases, the modifications made to further explain the infectious diseases of note (as a syndemic arrangement) are adequate, and compliment the paper's overall argument.